# Vertical integration of primary care practices with acute hospitals in England and Wales: why, how and so what? Findings from a qualitative, rapid evaluation

Manbinder Sidhu  ,[1] Jack Pollard,[2] Jon Sussex[3]

[1]Health Services Management Centre, University of Birmingham, Birmingham, UK
[2]Health Economics Research Centre (HERC), Nuffield Department of Population Health, University of Oxford, Oxford, UK
[3]RAND Europe, Cambridge, UK

**Correspondence to**
Mr Jon Sussex;
jsussex@randeurope.org

## ABSTRACT

**Objectives** To understand the rationale, implementation and early impact of vertical integration between primary care medical practices and the organisations running acute hospitals in the National Health Service in England and Wales.

**Design and setting** A qualitative, cross-comparative case study evaluation at two sites in England and one in Wales, consisting of interviews with stakeholders at the sites, alongside observations of strategic meetings and analysis of key documents.

**Results** We interviewed 52 stakeholders across the three sites in the second half of 2019 and observed four meetings from late 2019 to early 2020 (further observation was prevented by the onset of the COVID-19 pandemic). The single most important driver of vertical integration was found to be to maintain primary care local to where patients live and thereby manage demand pressure on acute hospital services, especially emergency care. The opportunities created by maintaining local primary care providers—to develop patient services in primary care settings and better integrate them with secondary care—were exploited to differing degrees across the sites. There were notable differences between sites in operational and management arrangements, and in organisational and clinical integration. Closer organisational integration was attributed to previous good relationships between primary and secondary care locally, and to historical planning and preparation towards integrated working across the local health economy. The net impact of vertical integration on health system costs is argued by local stakeholders to be beneficial.

**Conclusions** Vertical integration is a valuable option when primary care practices are at risk of closing, and may be a route to better integration of patient care. But it is not the only route and vertical integration is not attractive to all primary care physicians. A future evaluation of vertical integration is intended; of patients' experience and of the impact on secondary care service utilisation.

### Strengths and limitations of this study

► Primary care medical services in England and Wales are not usually operated by hospitals, but such vertical integration is now becoming more common and needs to be evaluated.
► This rapid evaluation starts to fill an evidence gap about the rationale for, and experience of, such vertical integration.
► We were able to conduct a large number of interviews (n=52) but had greater difficulty arranging non-participant observation of meetings (n=4).
► Due to the effects of the COVID-19 pandemic, the research team was unable to undertake the planned site-specific stakeholder workshops during data analysis.
► The study reported here provides initial findings and is intended to be complemented by a second evaluation.

including the UK National Health Service (NHS), in order to overcome fragmentation of services to patients and hence provide better quality care at the same or lower cost.[1] The evaluation reported here is of organisations running NHS acute hospitals that have also taken responsibility for running general medical practitioner (GP - primary care physician) practices. Integration between organisations operating at different stages along the patient care pathway is a form of 'vertical integration'.[1] The first instances of vertical integration between NHS acute hospitals and GP practices commenced in 2016 and the number, though modest, is growing. We aimed to find out more about precisely why and how vertical integration is being implemented in the NHS.

Examples of vertical integration are evident internationally. For example, in Valencia, Spain, a single contracted provider received

## INTRODUCTION

Better integration of healthcare across the organisations that serve patients has long been an objective of healthcare systems,

a fixed sum per capita to provide access to primary care, acute and specialist hospital services, although the contract has since been withdrawn.[2 3] In the USA, the Kaiser Permanente Community Health Initiative saw the establishment of an integrated healthcare delivery system, including both primary and secondary care, for 12 million members.[4] The Odense University Hospital cooperation model in Denmark introduced the colocation of primary and secondary care services, with an on-call GP facility located alongside the accident and emergency department.[5]

In England, acute hospitals are run by public organisations known as NHS Trusts and NHS Foundation Trusts (hereafter 'Trusts'), of which there are approximately 150. In Wales, acute hospitals are run by seven territorial Local Health Boards. UK residents register with a GP practice of their choice, usually near where they live. The practice receives an annual capitation fee for each person registered and provides primary care services and acts as a gatekeeper to secondary care, including acute hospital services. Traditionally, primary care has been provided by GP partners who own the GP practice and hold a General Medical Services contract with NHS England (in England) or with their Local Health Board (in Wales). In 2019 there were around 7000 GP practices in England, with over 34 500 GPs. In Wales there were slightly more than 400 practices and around 2000 GPs. However, the traditional partnership model of GP practice has been in decline and increasing numbers of GPs are salaried.[6] The demand for primary care has risen steadily in the UK[6] but in recent years the numbers of GPs has not kept pace.[6] NHS acute hospitals have also been facing cumulative annual increases in demand, especially from patients needing emergency care.[7] Thus the context for vertical integration is growing demand but stagnation in GP numbers.

Improving integration of patient care has been a focus of NHS policy concern for many years, with three major initiatives in the last decade. However, none of these initiatives entailed acute hospitals running GP practices.[8] In 2012, the English Department of Health established a programme of Integrated Care Pilots (ICPs) to explore different ways of providing health and social care services. An evaluation of the ICPs concluded that staff felt they worked more closely with other team members and improved communication with organisations across primary, secondary and community care. But patients did not report an awareness of improved care.[9] The ICPs were followed by two waves of integrated care and support 'pioneers', for which data on impact are still being collected.[8] The 2014 'NHS Five Year Forward View'[10] in England described options for greater integration between primary and secondary care, including Primary and Acute Care Systems, which would combine general practice and hospital services, although not within a single organisation. A report from an influential health policy body in England the following year encouraged similar integration and recommended

that acute hospitals take a greater role in primary care provision.[11]

Before 2019 most GP practices had formed collaborations of some kind with other GP practices.[12] The 2019 'NHS Long Term Plan'[13] included that all GP practices in England should form 'primary care networks' and since July 2019 this has been the case. But these primary care networks do not include hospitals. In parallel with these developments, the Welsh Government created 64 'primary care clusters' of GP practices.[14] Within these networks and clusters the GP practices remain as separate entities but coordinate for service planning and some other activities.

Developments in the NHS are not determined solely at the national level, however. There has always been locally led innovation and variation between local areas within the NHS in how services are organised. The most recent reorganisation of the NHS in England has created 42 geographically defined subnational groupings of providers and commissioners of care for NHS patients, called 'Integrated Care Systems'.[15] But these are structures for numerous organisations to work together, rather than new organisations providing integrated care, or joining hospitals and GP practices under common ownership. Vertical integration exists alongside and within Integrated Care Systems, but vertical integration is neither promoted nor created by Integrated Care Systems.

NHS organisations running acute hospitals additionally taking responsibility for managing GP practices has developed from local initiatives in a few places, alongside these nationally imposed primary care networks and clusters. It should be noted that vertical integration also entails an element of horizontal integration, as hospitals generally merge with more than one GP practice if any. GP practices in a vertically integrated organisation are also integrated horizontally with one another.

Theoretical literature on vertical integration between acute hospitals and primary care practices is limited.[16] This kind of vertical integration is a relatively new phenomenon, and much less explored, both theoretically and empirically, than the trend towards horizontal integration across primary care. Conrad and Dowling argued that vertical integration requires both administrative processes and clinical care to be integrated.[17] Shortell *et al* listed factors to be considered prior to commencing vertical integration, including trust among clinicians and institutions, having well-integrated information systems and non-clinical support services (back office functions), and consensus on practice and care delivery guidelines.[18] Ramsay and colleagues provide a typology of vertical integration: organisational, functional, service, clinical, normative and systemic.[19] The examples of vertical integration here mainly concern organisational integration (where organisations are brought together by mergers and/or structural change or through contracts between separate organisations) and functional integration (where non-clinical support and back office functions are integrated); along with some elements of service,

clinical and normative integration (where different clinical services are integrated at the organisational level and where patient care is integrated in a single process both within and across professions, eg, by use of shared guidelines). In our study we build on the Ramsay typology to examine capacity and governance, and whether vertical integration leads to stronger relationships between integrating organisations.

Overall, the reasons for vertical integration most commonly cited in the literature are expectations of it leading to better quality, more effective, healthcare, with better patient experiences of care, being delivered at the same or lower costs.

Empirical studies of vertical integration between hospitals and primary practices are scarce. Evaluations of Primary and Acute Care System pilot sites in England, which sought (prompted by the NHS Five Year Forward View[11]) to integrate care services, if not the organisations providing them, have not yet yielded clear conclusions on the outcomes. But there are signs that these and other models focused on care integration have had a beneficial impact on emergency admissions to hospitals,[20] and innovation has been observed in front-line services and system-wide collaboration.[12] However, findings are limited and questions exist over the reliability of outcome data.[21 22]

## METHODS

Against this background of rising demand for healthcare, GP shortages, policy that repeatedly refers to integrated care but that has hitherto prioritised horizontal integration, and a paucity of evidence in the literature, we conducted an evaluation of vertical integration with three explicit research questions: (1) what is the rationale for vertical integration; (2) whether and how vertical integration can underpin and drive the redesigning services offered in primary care settings; and (3) the impact on the general practice and hospital workforces. Our methodological approach was a rapid qualitative evaluation which included an iterative study design that was adapted to reflect the context of vertical integration in the UK, use of multiple qualitative data collection methods (interviews and non-participant observation), whereby data collection and analysis occurred in parallel by multidisciplinary researchers. The use of qualitative methods was suited to understanding the rationale behind the implementation of vertical integration and capturing the experiences of primary care and secondary care staff involved.

Vertical integration is a complex intervention in healthcare services. We have adopted the theory-driven approach to evaluation of complex interventions advocated by de Silva and colleagues, which is based on theory of change[23]; a framework which encourages development of comprehensive descriptions and illustrations of how and why a desired change is expected to happen in a particular context.[24] The approach advocated by de Silva *et al* was used as it provides a guide on how to construct a theory of change and how best to integrate its use into research projects seeing to evaluate complex interventions and/or phenomena. Applying theory of change methodology supports understanding of how stakeholders at each case study site interpret how vertical integration is making a difference to the local health economy. As part of our analysis, we developed a working theory of change and accompanying narratives for vertical integration specific to each case study site, as well as a generic model for cross-case comparison. Throughout, we have focused on mapping out or 'filling in' what has been described as the 'missing middle' between what a programme or change initiative does and how this leads to desired goals being achieved.[25] The full evaluation protocol is available at the National Institute for Health Research Journals Library.[26] A qualitative, cross-comparative, case study evaluation was undertaken. Interviews were conducted with stakeholders across three case study sites, alongside observations of strategic meetings and analysis of key documents, between summer 2019 and February 2020. The researchers—JS, MS and JP—are all experienced in conducting qualitative research with NHS managers and staff.

Five sites where vertical integration was present at scale and had been implemented for at least 2 years were initially identified across England and Wales through grey literature searching (in the journals: GP Online, Health Service Journal and Pulse). The selection of sites was purposive, seeking variation across case studies in terms of rural and urban geographies, and of the specific legal and governance frameworks used. Three sites were selected and approached, with each then providing written confirmation of participation. The sites are summarised in table 1.

| Table 1 | Case study sites | | | |
|---|---|---|---|---|
| **Site** | **Location** | **Legal framework** | **Date of implementation** | **No of GP practices vertically integrated** |
| 'Urbanville' | England; urban | Practices are part of hospital Trust and are sub-contracted to provide GP services | June 2016 | 10 |
| 'Greenvale' | England; rural | Creation of a subsidiary limited company managed by local trust | April 2016 | 13 |
| 'Seaview' | Wales; rural, coastal | Local health board direct control over primary and secondary care | March 2016 | 22 |

Semistructured interviews were undertaken in the second half of 2019, using a topic guide (see online supplemental file 1) to understand the rationale, drivers and challenges involved in the conceptualisation and implementation of vertical integration and interpret the experiences of primary and secondary care staff working together. Between 15 and 20 interviews were planned across each site (until no new themes were emerging from data). Participants were purposively sampled and approached through each case study site's gatekeeper.[27 28] The gatekeeper (a managerial stakeholder involved in the site-specific vertical integration set-up) facilitated the identification of key individuals involved in the design, implementation, governance and analysis of the model of vertical integration. Participants included: Trust and Local Health Board chief executives, NHS managerial-level staff and board members from service providers and commissioners; GPs and other primary care staff; and members of patient participation groups.

In late 2019 and early 2020 meetings were observed between key stakeholders at an executive and managerial level, to develop a better understanding of how decisions regarding implementation and delivery of vertical integration are made at local and executive board level. Public domain documents relating to the operation of the vertically integrated organisation, such as board minutes and presentations, were obtained from organisations' websites or from site gatekeepers and reviewed for all three sites. Learning from document analysis was triangulated with interview and observation during weekly team meetings and two half-day data analysis workshops.

Analysis of the qualitative data followed the seven stages of the Gale *et al* framework method for multidisciplinary health research.[29] All interviews were transcribed by a company specialising in transcribing health-related interviews, with a sample of the transcripts checked by MS. The three researchers familiarised themselves with the data by each reading all the transcripts from at least one site and at least two transcripts from each of the other two sites; and by holding an early data analysis meeting while interviews were ongoing. Once all the interviews had been completed, an inductive approach was applied to develop predefined codes, with two interview transcripts then independently coded by each study member in NVivo V.12 to ensure no important aspects of the data were missed. An analytical coding framework was then agreed on and reviewed at subsequent analysis meetings (see online supplemental file 2). The analytical framework was applied by indexing all the interview transcripts. The same codes were applied to notes from non-participant observations and to extracting data from the documentary review. A novel, rapid approach to charting codes was taken, with a matrix developed based on summaries of each code. The matrix was structured according to research questions and to facilitate the development of integrative themes. A further data analysis workshop was then held to finalise the development of themes. Write-up of the findings commenced after these were agreed. A summary of overall study findings was circulated to each site along with site-specific findings for their location, for factual checking.

### Patient and public involvement

The protocol for the study took account of input by a patient and public involvement member of the Birmingham, RAND, And Cambridge Evaluation (BRACE) Centre, Health and Care Panel, who reviewed a draft version. The BRACE Panel is made up of over 40 members, including eight patient and public members; system and organisational leaders; middle and operational clinical and general managers; frontline clinicians and other practitioner groups. Study findings were shared and discussed with the Panel, including patient and public members.

### RESULTS

The case study sites differed in their geographies—one urban ('Urbanville'), one rural with only one town with a population of over 25 000 ('Greenvale') and one rural/coastal with a handful of towns of that size ('Seaview')—and in how vertical integration was organised. In Urbanville, 10 GP practices (as at March 2020) are managed as a division of the acute hospital Trust, reporting directly to the Trust's board. In Greenvale, the hospital Trust has set up a limited company, which it 100% owns, to run its 13 (as at March 2020) GP practices. The company has its own board, which includes the Trust's chief executive and which in turn reports to the Trust board. The company runs its own back-office functions separately from the Trust. In Seaview, the Local Health Board that runs the local acute hospitals had, since March 2016, been directly managing GP practices that wished to give up their contracts—22 practices having done so as of March 2020.

We completed 52 interviews: 18 in Urbanville and 17 each in Greenvale and Seaview. Participant characteristics are shown in table 2. The programme of non-participant meeting observations had to be curtailed due to the arrival of the COVID-19 pandemic in March 2020 and the consequent unavailability of NHS staff. We completed four meeting observations before then: two in Greenvale and one each in Seaview and Urbanville. The large amount of interview data, supplemented by review of local documentation, plus the four observations that were completed, has yielded a rich source of qualitative data for analysis. The main results are reported here under the five major themes we identified.

### The main rationale for vertical integration is to maintain primary medical care local to where patients live

Literature suggested that better integration of patient care is a prime motive for vertical integration of primary care practices with acute hospitals. However, it soon emerged from the evaluation that the single most powerful driver of vertical integration at the case study sites has not been integration of patient care but, rather, the desire in the

**Table 2** Participant characteristics

| Area of specialism | Generic description of role | Number of participants |
|---|---|---|
| Primary care | Clinical | 12 |
| | Organisational management | 9 |
| | Professional representation | 2 |
| | Clinical and managerial | 1 |
| | Patient participation group | 2 |
| Primary and secondary care | Clinical | 1 |
| | Clinical and managerial | 1 |
| | Organisational management | 3 |
| Primary care commissioning | Organisational management | 1 |
| Secondary care | Clinical | 2 |
| | Organisational management | 15 |
| | Senior management | 3 |
| Total | | 52 |

face of GP shortages to maintain primary medical care local to where patients live:

It was quite apparent there was several practices that would fold, because – well, as you've heard a million times before – so elderly GPs having to retire, no one to take over the practice, or no desire to move it forward. Expensive locums to fill people, long-term vacancies. (Manager, Primary and Secondary Care)

there was that shared vision of 'How is this going to work? What is it going to look like?' We got that shared vision that primary care was…sinking. We were doing our best, but we just could not get on top of the amount of work we were being asked to do. (Manager, Secondary Care, Organisational Management)

Vertical integration has provided a more stable financial platform for primary care than individual practices being run as separate businesses. At the case study sites, the business risks associated with running a general practice have been removed from the GPs, who no longer risk personal financial loss when the practice suffers from high costs, for example, due to employing locum doctors. Those risks have been absorbed by the organisation running acute hospitals in the area. Trusts and Local Health Boards have annual budgets of hundreds of millions of GB pounds, much larger not only than those of individual GP practices (annual budgets typically under £5 million) but also of primary care networks and clusters (which anyway do not generally pool the budgets of their participating practices). Consequently, a Trust (England) or Local Health Board (Wales) is better

able than a GP practice to cope with the financial risks. At both Greenvale and Urbanville, public domain documents stated the view that the respective hospital trusts considered that overall they were financially better off having vertically integrated with the GP practices, once account was taken of the better management of where patients were treated. Yu and colleagues estimated that at one vertical integration site in England estimated savings due to reductions in unplanned care attributed to vertical integration amounted to £1.7 million annually.[30]

If you just measure primary care on their contract value, which actually in comparison to how much it would cost to run an acute trust is minuscule, you know, is a fraction of the cost, but if you look at their impact and influence on spend in the system, that is, how many referrals they make, how many they signpost, how many they recommend someone to go to residential nursing care, how many they refer, you know, into other services, to how much they try, you know, how much testing they do, their costs are something like seven times what their contract value is. (Manager, Primary Care, Organisational Management)

The Trust-backed or Local Health Board-backed GP practices can also offer more staff training and career development opportunities, and greater job security, which increases the chances of recruiting and retaining primary care staff. For the Trust or Local Health Board, keeping GP practices open makes it easier to manage the flow of patients to the acute hospital and enable patients to be cared for locally when that is more appropriate.

We heard the expectations of interviewees about the future interaction of vertical integration with horizontally integrated primary care networks (England) or primary care clusters (Wales). At Urbanville (England), all but one of the vertically integrated GP practices together formed a single, large network. The one other vertically integrated practice was part of a different primary care network with a majority of non-vertical integration practices. Thus, with this one exception, the primary care network was coterminous with the vertical integration organisation. The interviewees at Greenvale (England) who offered views on the future interaction of the vertical integration company with the local primary care networks thought that the two forms of integration could coexist.

'when you get to a point where you're trying to support practices that get into big financial difficulties, the PCN isn't the risk-holding entity. So the question then is: Who holds the financial liability for that practice? So I think the PCN will change the nature of the relationship with [the vertical integration company], and it will almost put a step in place before you get to an entity like [the vertical integration company], that scale provider of local practices trying to support a practice that is in difficulty and kind of keep it stable. But there's likely to still be a requirement for an

at-scale provider like [the vertical integration company], who can then come in. (Manager, Secondary Care, Organisational Management)

The emphasis at Seaview (Wales) was on stabilising GP practices in the hope of later returning them to independent operation, and thus vertical integration is there seen as a temporary state, although one that may continue with a changing group of practices within the vertically integrated arrangement as new practices join and others leave it. There is no sign of vertical integration coming to an end in Seaview, and few practices that have been vertically integrated have subsequently returned to independent operation.

### Vertical integration depends on previous good relationships between primary and secondary care providers

We found that closer organisational integration could be attributed to previous good relationships between primary and secondary care locally, and to there being a history of planning and preparation for integrated working. Vertical integration at Greenvale was facilitated, at least partly, by the Primary and Acute Care System model of care that had been operating in that area since 2015, and which focused on better managing care across primary and secondary care settings for patients with complex and multiple morbidities.

The structural divide in the NHS between GP practices delivering primary care services and Trusts or Local Health Boards running hospitals has not been fully overcome. Many local GP practices choose to remain outside the vertical integration arrangement even though they would be free to join it. Occasionally this may be due to suspicions among some GPs about the motives of hospitals offering vertical integration, although we did not find this to be a prevalent belief:

I think there's still a lot of negative sentiment and a belief that [the vertical integration company] was set up as a Trojan horse to kind of take over general practice… (Commissioning Manager, Organisational Management)

More commonly expressed was the view that GP practices that were financially stable and had no great recruitment problems were just not that interested in vertical integration.

### The transition from independent GP practices to salaried providers of primary care within a vertically integrated organisation was not always smooth

An unintended consequence of the transition to vertical integration may have been that some individual GPs left their practices sooner than they might otherwise have done–because the vertical integration meant that they could exit without financial cost to themselves and with confidence that the practice would remain open for local patients. The transition from being GP partners to salaried doctors within a vertically integrated organisation was understood by some of the GP partners viewing a salaried

employee position as a temporary state. They remained for only a short period of time post-vertical integration and then left general practice:

When we took on that practice, we knew they were in trouble because they'd tried to recruit for four years and they hadn't been able to. There were three partners there and a salaried GP, and, at the time, we had assurances from everyone, including the CCG, that those partners would stay if (the vertical integration company) took them on and helped stabilise the practice. Within…a week of taking them on, the salaried GP retired, and then one of the partners resigned, and then subsequently the other two partners both went off sick. (Manager, Primary Care, Organisational Management)

Practice staff who moved into vertically integrated organisations had their terms and conditions or employment protected. This resulted in more job security. But they also experienced greater scrutiny with regard to job specifications and whether they fulfilled them. The move to vertical integration imposed a significant requirement on hospital Trust and Local Health Board staff, used to operating in large organisations focused on secondary care, to learn about and understand the practicalities and the culture of running primary care. Primary care and secondary care continue to be seen as fundamentally different to operate:

They (Trust staff) didn't really understand primary care…they're jumping to something, but they didn't really know how to run it. (Primary Care Clinician, Primary Care)

I think the balance in secondary care – very management heavy. General practice – very clinical leadership heavy. And the realignment of that, I think, is a real challenge. (Senior Manager, Primary and Secondary Care, Organisational Management)

### Vertical integration facilitated changes to patient care but may not have driven them

Although we found that changing patient care was not the prime motive for vertical integration, the platform it created by stabilising primary care provided the opportunity to innovate. Service innovations were evident in all three case studies. But it was hard to tell the extent to which the changes, such as specialist musculoskeletal or diabetic services being provided at some GP practices, might have occurred anyway in the absence of vertical integration. As already noted, attention was predominantly on keeping GP practices going; and vertical integration was enabled by the prior existence of good relationships between the acute hospital and local GP practices, so that vertical integration was described more as a consequence than a cause of such good relationships. No major changes in relationships between GPs and hospital specialists were identified that could confidently be attributed to vertical integration.

It was clear, however, that without financially stable, fully staffed primary care practices, service innovations would have been harder to introduce. Among innovations introduced by the vertically integrated organisations were: sharing of information in real time across primary and secondary care (Urbanville); and targeting high-risk patients with multiple morbidities who are most likely to access emergency secondary care but could be better managed in the community (Greenvale).

> It's like a dashboard, so it tells us what patients have been admitted overnight, what procedures they've had done, and we get a copy of that each day and the GPs look at it and think, 'Oh, Mrs so-and-so was in there last night; I might give her a call and see if she's OK.' So it links the care up better. (Manager, Secondary Care, Organisational Management)

However, at the meetings we observed, the discussions focused mainly on the operation of primary care, with comparatively little reference to interaction between GP practices and the acute hospital.

### Vertically integrated practices are attractive to some (potential) staff but recruitment of GPs is still not easy

All three sites had some success in recruiting salaried GPs to work within vertical integration practices. The reduction in personal financial risk for GP partners that results from the Trust or Local Health Board taking responsibility for the GP contract, seems to have helped. Combined with increased training and opportunities for GPs to develop specialist interests, the opportunity for GPs to focus on clinical work and leave the 'running of the business' to others makes vertically integrated practices more attractive to some potential GP recruits. But recruitment of GPs is not easy even for vertical integration organisations, and all sites continued to encounter high costs associated with employing locums.

The vertical integration sites were able to increase the use of multidisciplinary teams in primary care. There were increased training opportunities for non-clinical staff in primary care to upskill and 'move up' within a larger organisation, which may have improved their recruitment and retention within the vertical integration model:

> There are several practices that offer apprenticeships, and an external person joined the meeting to present about the potential opportunities for [the vertically integrated organisation in Greenvale] to provide further apprenticeships. (Observation, Practice Managers meeting)

### DISCUSSION

This qualitative evaluation of vertical integration of acute hospitals with primary care practices in England and Wales has found that the most powerful rationale was to maintain primary care locally when GP practices might have closed. In this way, vertical integration has preserved local access to primary care for those practices registered populations and helped with continued management of patient flows to hospital. The platform afforded by that enables service innovations to be introduced, although similar innovations are being introduced in primary care beyond the vertically integrated practices. The transition from independent primary care practices, mostly run as GP partnerships, to being parts of larger organisations employing salaried staff, has had some positive workforce impacts including on recruitment, but longstanding cultural differences between primary care and secondary care staff remain.

The study team completed a qualitative rapid mixed method comparative evaluation, following established methodology and guided by previous evidence of implementation.[31 32] To our knowledge, this is the first empirical study investigating the rationale and implementation of vertical integration in the UK. A follow-up evaluation to measure impact and effectiveness is planned. Data were closely analysed and reviewed in frequent discussions among the research team, including two analysis workshops. The large number of interviews (n=52) gives confidence that our findings are robust.

Across all three sites, fewer than intended non-participant observations were completed due to meetings being rescheduled/cancelled at short notice as a result of the onset of the COVID-19 pandemic. Access to such data would have further strengthened the robustness of our findings, as well as potentially giving the study team increased access to a wider number of stakeholders to approach for interview, rather than being reliant solely on our gatekeepers. The success of interviewee recruitment likely depended on the strength of the relationship between the gatekeeper and the interviewee. This may be a source of bias in the sample of interviews obtained. Nevertheless, our interviews included people outside the vertically integrated organisations whose views thus provided a different perspective.

Notwithstanding contacting a range of stakeholders at each site, a small number of stakeholders, predominantly hospital staff, did not respond to our invitations. However, from secondary care staff interviews that we were able to conduct, the team does not believe this imbalance across primary and secondary participants influenced the overall data set, but rather focused our interpretation on the most pertinent purpose for vertical integration, that is, creating stability and sustainability in primary care.

Despite data collection instruments developed to ascertain information about the financial set-up at each site, data on this topic were scarce. So too was information about how patient experience may have changed, for better or worse. We intend to address major aspects of the financial impact on the local healthcare system of vertical integration, and of the experience of patients (particularly, those considered to be high users of healthcare services, for example, those living with multiple long-term conditions), in a follow-up mixed-methods evaluation

including analysis of hospital activity data and interviews with patients.

Empirical evidence on the rationale, implementation and impact of vertical integration between acute hospitals and primary care practices, is thin. Our findings differ from the rationale for vertical integration most commonly discussed in previous literature, which suggested that the main driver would be to improve the quality of patient care and patient experience through better coordination between different parts of the care pathway. In the context of the NHS in England and Wales in 2019, we found that the main rationale was more basic: to sustain primary care practices.

Hence, our evaluation implies that vertical integration where the organisation running an acute hospital takes over the management of primary care practices is a valuable option to consider when those practices look likely to fail. By sustaining primary care in such circumstances, vertical integration provides continued local access for patients to that care and creates opportunities for better integration of care. But not all primary care staff favour integration with a hospital, and it is not something that should be imposed from the top down.

Our qualitative evaluation based on three case studies has found much of interest, but important questions remain. In particular, research is needed to examine outcomes and impacts, including for patient and carer experience, associated with hospitals managing GP services, and to explore the impact of vertical integration on overall local healthcare system costs. Any future evaluation of vertical integration will take place in the context of the great changes to primary care practice that have come about as a result of the COVID-19 pandemic. A longer-term assessment of the impact of vertical integration in a world where telephone and digital consultations in primary care have become more common than face-to-face consultations, could represent an important body of work for the wider research community in future.

**Acknowledgements** We thank Richard Allen and Nishma Manek for advising on our study protocol from a patient and public involvement (PPI) and GP perspective; and Professor Russell Mannion, Professor Judith Smith, Dr Joanna Ellins and Dr Iestyn Williams (all University of Birmingham) and Professor Nicholas Mays (London School of Hygiene and Tropical Medicine) for their advice to the authors.

**Contributors** MS, JP and JS were responsible for the study conception, design and drafting of the manuscript. MS, JP and JS completed data collection and MS and JS performed data analysis. MS, JP and JS wrote the paper. JS was principal investigator for the study and is guarantor for the content of this article.

**Funding** This project was funded by part of a grant from The National Institute for Health Research, Health Services and Delivery Research programme (HSDR 16/138/31 – Birmingham, RAND and Cambridge Evaluation Centre).

**Competing interests** None declared.

**Patient consent for publication** No patients were approached to participate in this evaluation.

**Ethics approval** This study involves human participants. The University of Birmingham's Research Ethics Committee provided ethical approval in July 2019 (ERN_13-1085AP35). Participants gave informed consent to participate in the study before taking part.

**Provenance and peer review** Not commissioned; externally peer reviewed.

**Data availability statement** Data sharing not applicable as no datasets generated and/or analysed for this study. Due to the consent process for data collection at case study sites within this evaluation, there are no data that can be shared.

**ORCID iD**
Manbinder Sidhu http://orcid.org/0000-0001-5663-107X

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
