## [Reviewer comments · BMJ Open]

ARTICLE DETAILS

TITLE (PROVISIONAL)	The vertical integration of primary care practices with acute hospitals in England and Wales: why, how and so what? Findings from a qualitative, rapid evaluation
AUTHORS	Sidhu, Manbinder; Pollard, Jack; Sussex, Jon

VERSION 1 – REVIEW

REVIEWER	Georgia Black University College London, Applied Health Research
REVIEW RETURNED	23-Jun-2021

GENERAL COMMENTS	Thank you for the opportunity to review this manuscript. The topic of the research is important, and I think a real strength of the approach is the comparison between three sites. I understand that the pandemic has limited the amount of observation that the authors were able to do. However, I had real difficulty getting the gist of what the article was about and how it was being done. These are major concerns which require substantial changes if the article is to be published. I will outline my concerns by section. INTRO The precis of integrated care is fairly comprehensive, and takes a historical approach. However, there are some glaring omissions such as the Integrated Care Pilots and their conclusions. I'm not sure that I agree with your conclusions that "Theoretical literature on vertical integration between acute hospitals and primary care practices is limited", and would urge you instead to unpick what new direction(s) you are taking, and why. For example, how you might build on the Ramsay typology, or why you might critique Shortell et al. I also suggest that you talk a bit about the devolution agenda and Integrated Care Systems as an important new vehicle for integration (particularly given your conclusions about the vulnerability of primary care). METHODS The level of detail here is rather scarce. It's not clear whether you are including the scoping review as part of your empirical data here - there's no detail about the methods or methodology chosen at all. - please say why you chose de Silva approach to evaluation of complex interventions, and also explain for the reader what theory of change means (and why it applies here).- please also include detail on how this affected your analytic strategy e.g. was it part of your analytical framework- it's not clear how you integrated your review findings with your
---

	interview and observation data FINDINGS The thematic structure is confusing - the theme titles don't really reflect what the content says and there is relatively little quotation data to support the conclusions. There's also no observation data at all. If your aims had been clearer, it would have perhaps made it easier to structure the findings. The clearest aim is around the rationale for integration. If your whole paper was geared around this, it would be much more focussed. I don't get a sense of the 'how' in your title at all. In general, the findings are rather impoverished and lack detail, illustrative examples and any sense of chronology. DISCUSSION I agree with your summary in the Discussion that your findings are really about the difference between the stated and real aims of vertical integration. Your strengths and limitations section could be a bit stronger.
--	---

REVIEWER	Rebecca Rosen Nuffield Trust
REVIEW RETURNED	15-Jul-2021

GENERAL COMMENTS	This paper is topical and addresses an area of change in the organisation of primary care services that should be of interest to policy makers and practitioners and is currently under-researched. The study design is for the most part appropriate for the objectives with one possible exception (see below relating to data on financial implications). The paper is well structured, clearly written, describes methods and analysis well and provides a good, rich summary of data on the rationale for vertical integration. I think there are two weaknesses in the paper - one of which could be addressed with minor revisions and the other probably reflects a missed opportunity during data collection. The first weakness is in the balance between different sections of the paper and the limited presentation of data on changes to patient care and impact on staff. There is passing reference to changes in MSK and diabetes services (including acknowledgement that these changes were not necessarily as a result of vertical integration) but I want to know whether the new link between practices and the acute hospital led to a different model of change compared to pre-integration. Were there any changes in relationships with specialists? Did the GPs interviewed notice / report any difference in the way they worked with these services? Did the hospital clinicians perceive any new opportunities for redesigning after integration - even if they have not yet been implemented? The message that the main rationale for VI was to maintain local primary care is very clear. It would be good to have some deeper insights into these other issues. The introduction and methods sections are currently long and could perhaps be edited down to make room for these if additional data are available. The second area of weakness relates to the financial context for
---

	integration presented in lines 45 - 55. The authors report that the practices have become more financially stable and that financial risks can be absorbed into trust finances of hundreds of millions of pounds. Further insights about the level of investment needed to stabilise the practices would have been very useful. Was there any information about this in Trust Board papers? Financial accounts or any other documents analysed for the study? Is there additional interview data which has not been presented due to lack of space? The extent to which hospitals which vertically integrate with general practice provide financial subsidies to the practices is important. Policy makers looking for ways to strengthen integration between primary and acute services may see the stabilisation of practices as a worthwhile end in itself but it's important to know at what cost. The primary objective of the study is to explore the rationale for vertical integration but it does also seek to examine early impacts. The limited data on financial impacts should probably not prevent publication but if there is additional data available it would be very useful to include it. Overall, this well-written paper makes a useful contribution to the literature on primary care integration. It could be improved with minor revisions and additional data on impact on services delivery, staff and financial flows if this can be found in Trust documents/ is available
--	---

VERSION 1 – AUTHOR RESPONSE

Reviewer 1	Comments (Dr. Georgia Black, University College London)	Response
1	Thank you for the opportunity to review this manuscript. The topic of the research is important, and I think a real strength of the approach is the comparison between three sites. I understand that the pandemic has limited the amount of observation that the authors were able to do. However, I had real difficulty getting the gist of what the article was about and how it was being done. These are major concerns which require substantial changes if the article is to be published.	N/A.
2	INTRO The precis of integrated care is fairly comprehensive, and takes a historical approach. However, there are some glaring omissions such as the Integrated Care Pilots and their conclusions. I'm not sure that I agree with your conclusions that "Theoretical literature on vertical integration between acute hospitals and primary care practices is limited", and would urge you instead to unpick what new	We have included reference to the Integrated Care Pilots and their conclusions. We have explained that vertical integration between acute hospitals and GP practices remains a relatively new phenomenon, and much less explored, both theoretically and empirically, than

	direction(s) you are taking, and why. For example, how you might build on the Ramsay typology, or why you might critique Shortell et al.	the trend towards horizontal integration across primary care. We have added how we have built on the Ramsay et al. typology.
3	I also suggest that you talk a bit about the devolution agenda and Integrated Care Systems as an important new vehicle for integration (particularly given your conclusions about the vulnerability of primary care).	We have referred to the devolution agenda and the onset of Integrated Care Systems. There have been many previous reorganisations of the NHS and it is much too soon to tell whether 'Integrated Care Systems' in England will prove to be vehicles for better integration. But we have added text so that the reader is made aware of the 'Integrated Care Systems' policy as part of the context, but not a driver, of vertical integration between acute hospitals and GP practices.
4	METHODS The level of detail here is rather scarce. It's not clear whether you are including the scoping review as part of your empirical data here - there's no detail about the methods or methodology chosen at all.	We are not including the scoping review as part of this paper, and have edited the wording in the results when referring to relevant literature. We have included a statement about our choice of methodology and methods.
5	please say why you chose de Silva approach to evaluation of complex interventions, and also explain for the reader what theory of change means (and why it applies here).	We have added a justification for choosing the de Silva et al. approach and what a theory of change means and why it applies to our study.
6	please also include detail on how this affected your analytic strategy e.g. was it part of your analytical framework	We have included further detail on how our theoretical approach was embedded in the analysis.
7	it's not clear how you integrated your review findings with your interview and observation data	We have clarified that analysis of public domain documents was triangulated with interview and observation data

		during team meetings and data analysis workshops.
8	The thematic structure is confusing - the theme titles don't really reflect what the content says and there is relatively little quotation data to support the conclusions. There's also no observation data at all.	We have changed the theme titles to more directly reflect the content. We have included additional quotes and observation data (where relevant) to support our presentation of findings.
9	If your aims had been clearer, it would have perhaps made it easier to structure the findings. The clearest aim is around the rationale for integration. If your whole paper was geared around this, it would be much more focussed. I don't get a sense of the 'how' in your title at all.	We have revised the text to make the aims of the study clearer: in the Objectives in the Abstract; in the overall aim of the evaluation stated in the first paragraph of the Introduction; and in more detail as part of the Methods section. As part of our study, our interviewees were much more forthcoming about the rationale of vertical integration and its purpose, but less so with regard to the "how" and even less concerning the impact. Our manuscript and subsequent data also reflect the stage of the transition to vertical integration that each site was at when we completed data collection, i.e. relatively early on, where evidence of impact was limited.
10	In general, the findings are rather impoverished and lack detail, illustrative examples and any sense of chronology.	We have included additional quotes and refer to further illustrative examples to add detail and have added text about timescales and sequencing. Given the small number of sites undertaking vertical integration in the UK, we are constrained, in order to protect the anonymity of sites and participants, not to provide all details and chronologies.

11	DISCUSSION I agree with your summary in the Discussion that your findings are really about the difference between the stated and real aims of vertical integration.	N/A.
12	Your strengths and limitations section could be a bit stronger.	We have added to our strengths and limitations section.
Reviewer 2	Comments (Dr. Rebecca Rosen, Nuffield Trust)	Response.
13	This paper is topical and addresses an area of change in the organisation of primary care services that should be of interest to policy makers and practitioners and is currently under-researched.	N/A.
14	The study design is for the most part appropriate for the objectives with one possible exception (see below relating to data on financial implications). The paper is well structured, clearly written, describes methods and analysis well and provides a good, rich summary of data on the rationale for vertical integration.	N/A
15	I think there are two weaknesses in the paper - one of which could be addressed with minor revisions and the other probably reflects a missed opportunity during data collection. The first weakness is in the balance between different sections of the paper and the limited presentation of data on changes to patient care and impact on staff. There is passing reference to changes in MSK and diabetes services (including acknowledgement that these changes were not necessarily as a result of vertical integration) but I want to know whether the new link between practices and the acute hospital led to a different model of change compared to pre-integration. Were there any changes in relationships with specialists? Did	Interviews, meeting observations and document review did not reveal whether the new link between practices and the acute hospital led to a different model of change compared to pre-integration. Attention was predominantly on keeping GP practices going. No major changes in relationships with specialists were identified that could confidently be attributed to vertical integration. We have added text to make this clear.

	the GPs interviewed notice / report any difference in the way they worked with these services? Did the hospital clinicians perceive any new opportunities for redesigning after integration - even if they have not yet been implemented?	
16	The message that the main rationale for VI was to maintain local primary care is very clear. It would be good to have some deeper insights into these other issues. The introduction and methods sections are currently long and could perhaps be edited down to make room for these if additional data are available.	We have added quotes to enhance the insights. We have added other text to explain the difficulty interviewees had in attributing other consequences to vertical integration per se.
17	The second area of weakness relates to the financial context for integration presented in lines 45 - 55. The authors report that the practices have become more financially stable and that financial risks can be absorbed into trust finances of hundreds of millions of pounds. Further insights about the level of investment needed to stabilise the practices would have been very useful. Was there any information about this in Trust Board papers? Financial accounts or any other documents analysed for the study? Is there additional interview data which has not been presented due to lack of space? The extent to which hospitals which vertically integrate with general practice provide financial subsidies to the practices is important. Policy makers looking for ways to strengthen integration between primary and acute services may see the stabilisation of practices as a worthwhile end in itself but it's important to know at what cost. The primary objective of the study is to explore the rationale for vertical integration but it does also seek to examine early impacts. The limited data on financial impacts should probably not prevent publication but if there is additional data available it would be very useful to include it.	We have added text about financial impacts of VI as seen by interviewees, participants of meetings we observed, and public domain documentation (which we cannot reference due to the need to maintain anonymity). We have also added a reference from the literature (Yu et al. 2020) that estimates the net financial impact of vertical integration at a location in England they have analysed.
18	Overall, this well-written paper makes a useful contribution to the literature on primary care integration. It could be improved with minor revisions and additional data on impact on	See response to our prior comments.

	services delivery, staff and financial flows if this can be found in Trust documents/ is available	
--	--	--

VERSION 2 – REVIEW

REVIEWER	Georgia Black University College London, Applied Health Research
REVIEW RETURNED	04-Dec-2021

GENERAL COMMENTS	Thank you for another chance to read this interesting and important paper. Since the last revision, the authors have made some substantial changes to the manuscript, and it is much clearer as a result. In particular, I like the new theme titles which give the reader a rich story of the findings. I think the findings are generalisable to other integrated services that I have witnessed, and I hope this paper makes an impact on implementation and improvement of these collaborative organisations. I think it's ready for publication but would suggest a thorough proof-reading first.
---

REVIEWER	Rebecca Rosen Nuffield Trust
REVIEW RETURNED	24-Nov-2021

GENERAL COMMENTS	Thanks for asking me to re-read the revised version of this paper. The additional data in this version contribute to a richer and more balance account of vertical integration – particularly in illustrating the financial and workforce risks assumed by the acute trusts /local health board when taking on GP practices. Also, in painting a more detailed picture of the motivations for integration. Although the revisions and additional data take the paper over the usual 4000 word limit, it is clearly written and flows well so it remains easy and engaging to read. If the editor wants the paper slightly shorter, I'm not convinced the new paragraph about ICS formation on page 4 adds much. And while the additional quotes throughout add value, a few could be shed – specifically on pages 11 and 12. There are a few typos scattered through the paper. On page 5 first para line 10 the authors say they will build on Ramsay's typology to examine 'improved capacity and governance' which seems to assume from the start that capacity will improve. Should it just read as 'capacity and governance'? I recommend that this revised version of the paper be accepted for publication.
--

VERSION 2 – AUTHOR RESPONSE

Reviewer	Comments (Dr. Georgia Black, University College London)	Response
1		

1.	Thank you for another chance to read this interesting and important paper. Since the last revision, the authors have made some substantial changes to the manuscript, and it is much clearer as a result. In particular, I like the new theme titles which give the reader a rich story of the findings. I think the findings are generalisable to other integrated services that I have witnessed, and I hope this paper makes an impact on implementation and improvement of these collaborative organisations. I think it's ready for publication but would suggest a thorough proof-reading first.	Thank you for your comments. The authors have proofread the manuscript prior to submission.
Reviewer 2	Comments (Dr. Rebecca Rosen, Nuffield Trust)	Response.
2.	Thanks for asking me to re-read the revised version of this paper. The additional data in this version contribute to a richer and more balance account of vertical integration – particularly in illustrating the financial and workforce risks assumed by the acute trusts /local health board when taking on GP practices. Also, in painting a more detailed picture of the motivations for integration.	N/A.
	Although the revisions and additional data take the paper over the usual 4000 word limit, it is clearly written and flows well so it remains easy and engaging to read. If the editor wants the paper slightly shorter, I'm not convinced the new paragraph about ICS formation on page 4 adds much. And while the additional quotes throughout add value, a few could be shed – specifically on pages 11 and 12.	The additional paragraph on ICS formation was included in response to Reviewer 1 comments; therefore, we have kept this paragraph in our introduction. We have omitted a select number of quotes on pages 11 and 12.
	There are a few typos scattered through the paper. On page 5 first para line 10 the authors say they will build on Ramsay's typology to examine 'improved capacity and governance'	We have proofread the manuscript and corrected any typos found.

	which seems to assume from the start that capacity will improve. Should it just read as 'capacity and governance'?	We have edited the sentence to read 'capacity and governance'.
	I recommend that this revised version of the paper be accepted for publication.	Thank you.